# OpenReview forum: "Fake it till You Make it: Reward Modeling as Discriminative Prediction"
_ICLR.cc/2026/Conference — Submitted to ICLR 2026_

### Official Review · Reviewer_anJG · 2025-10-20

**Soundness:** 2
**Presentation:** 3
**Contribution:** 3
**Rating:** 2
**Confidence:** 4

**Summary:**

GAN-RM is proposed as an efficient reward modeling framework for aligning visual generative models with human preferences. Inspired by adversarial training in GANs, it addresses limitations of existing reward models, which rely on expensive human-annotated preference data or manually engineered quality metrics that may not fully capture preferences. GAN-RM trains a discriminator to distinguish a small set of unpaired, representative from outputs of the generative model. This eliminates the need for pairwise annotations.

**Strengths:**

The paper circumvents the need for human annotators for fine-tuning and draws inspiration from classical adversarial training of GANs for diffusion models.

**Weaknesses:**

- The paper does not discuss stability concerns typical of GAN/adversarial training. Cross-entropy loss probably avoids these known issues. Can the authors more explicitly discuss this as part of the paper?
- Is this really a novel method? It seems the authors simply use the discriminator and then leverage it either to do test time scaling, SFT, or DPO. But they dont really propose a new algorithm?

The method is addressing a different topic but it would be good to contextualize it with other works mixing diffusion and adversarial training in other setups: Diffusion Adversarial Post-Training for One-Step Video Generation, Adversarial Diffusion Distillation, etc

**Questions:**

Are you using LPIPs for video at the frame level?

Why does performance plateau so quickly with data size in the ablations any insights? I wonder what happens with thousands or tens of thousands of images that are more common in DPO and would make for a more realistic comparison. Standard DPO generally performs better with more preference data so you may be compering the methods in a regime that is bad for DPO.

---

### Official Review · Reviewer_4f9p · 2025-10-27

**Soundness:** 2
**Presentation:** 2
**Contribution:** 1
**Rating:** 4
**Confidence:** 3

**Summary:**

This paper proposes GAN-RM, a reward modeling framework for aligning visual generative models with human preferences. The method trains a discriminator (frozen CLIP encoder + trainable MLP) to distinguish between a small set of "Preference Proxy Data" (representative high-quality samples) and model-generated outputs. The discriminator's confidence score serves as the reward signal for downstream applications including Best-of-N sampling, Supervised Fine-Tuning, and Direct Preference Optimization. The authors evaluate on image quality (SD1.5, SDXL), safety alignment, and video generation (VideoCrafter2). For image quality, they use 500 samples from JourneyDB as proxy data and for safety, 15,690 samples from CoProV2. The method supports multi-round iterative training where both reward model and generator are updated. The authors claim comparable performance to DiffusionDPO (trained on 1M annotated pairs) while using only hundreds of samples, though the safety experiments actually use over 15K samples.

**Strengths:**

1. The method presented by the paper is computationally efficient by freezing the CLIP encoder and only training a small MLP head. This makes training fast and practical for real-world deployment.
2. Broad experimental validation across multiple base models (SD1.5, SDXL, VideoCrafter2), multiple domains (image quality, safety, video), and multiple post-training methods. The method demonstrates consistent improvements across diverse settings.
3. Comprehensive ablation studies is presented in the paper. The supplementary material provides ablation on PPD size, image quality and diversity, hyperparameter sensitivity.
4. Human validation provided through user study showing 74.4% preference over base case model.

**Weaknesses:**

1. The core claim of "eliminates manual preference annotation" is misleading. The method still depends on a pre-curated Preference Proxy Data (PPD) of 500 high-quality JourneyDB images, i.e. someone already judged quality. And comparing 500 curated samples against 1M crowd-sourced labels are not fair.
2. The paper has limited technical novelty. "GAN" is mostly branding. The reward model is a CLIP-vision feature extractor + small MLP trained as a binary classifier to separate PPD images from model outputs. Only the MLP is updated, there is no adversarial generator training loop.
3. The data requirements are inconsistent across experiments. The paper states using only 500 samples, but safety experiments actually use 15,690 samples. This raises questions about true data requirements for different applications.
4. Hyperparameter effects are small, suggesting design choices are not critical. In the sensitivity study, FID varies only modestly across $\alpha$ and $M$ (e.g. 14.18 - 14.45).
5. Computational cost reporting lacks comparative analysis. The paper lists hardware and clock time but provides no cost or comparison vs baselines leaving the practical efficiency trade-offs unclear.

**Questions:**

1. The author state that the method "eliminates manual preference annotation", yet the PPD relies on pre-calculated high-quality JourneyDB images. How do you reconcile this? Doesn't the curation burden simply shift to selecting a high-quality proxy set?
2. Why does safety alignment use ~15k samples while quality uses ~500? What guidance can you give users for choosing PPD size across tasks with different objectives and risks?
3. Have you tested intentionally biased or non-representative PPD? What failure modes emerge, and what diagnostics or safeguard do you recommend?

---

### Official Review · Reviewer_LcJD · 2025-10-30

**Soundness:** 3
**Presentation:** 4
**Contribution:** 3
**Rating:** 4
**Confidence:** 5

**Summary:**

The paper introduces GAN-RM, a practical framework for training an effective discriminator, trained to distinguish a small set of user-provided high-quality Preference Proxy Data (PPD) from model outputs. This discriminator is built upon open-source, pretrained models (CLIP-Vision), with a small MLP called RPL. Using this discriminator, multiple methods are used (Best-of-N, SFT, DPO) to improve the generation quality, steering them towards the Preference Proxy Data. The method is simple and data-efficient (the authors report competitive gains compared to DiffusionDPO using 1M labeled pairs while GAN-RM uses ~500 proxy images) and is validated on several image/video models (SD1.5, SDXL, VideoCrafter2).

**Strengths:**

1. This discriminator is built upon open-source, pretrained models (CLIP-Vision), with a small MLP called RPL, making it accessible to all practitioners; the results are easily reproducible.
2. The method is indeed data-efficient, using a very small and unpaired dataset to train the discriminator.
3. Strong empirical validation, extensive experiments show that this simple method can indeed be somewhat equivalent to much more expensive methods like DiffusionDPO.

**Weaknesses:**

Major concerns:

1. The name GAN-RM is conceptually misleading: there is no adversarial training between generator and discriminator, nor any min–max optimization. The method consists of supervised discriminative training of a reward model (binary classifier) followed by generator fine-tuning using pseudo-preference data. It would be clearer to present this as Reward Modeling via Discriminative Prediction rather than a GAN variant. The current terminology may confuse readers and overstate the conceptual novelty.
2. The reward model (discriminator) appears to take only image inputs, trained to distinguish PPD vs generated samples. In a text-to-image (or text-to-video) setting, preference is inherently prompt-dependent. CLIP-Score and the other mentioned reward models are all conditioned on text. Ignoring text-conditioning means the reward cannot capture semantic faithfulness, only visual or stylistic quality. As far as I can see, the discriminator only relies on visual information, and no text-conditioning is employed. This is a fundamental limitation not adequately addressed.
3. The rank-based label assignment (y = e^{-α·r}) is empirically motivated but lacks grounding. It’s unclear why exponential decay is appropriate, or whether it introduces bias.
4. We are not even doing RL, and the task is not related to any RL methodology, beyond the shared spirit with RLHF techniques; why do we call this a reward-modelling scheme, rather than a data rating scheme?

Minor concerns:

1. The first two paragraphs could use more citations to strengthen the claims.
2. The paper from K. Black et. al. [1] definitely deserves a citation.
3. Many sections from the related works section, especially 2.1, are conceptually irrelevant to the paper.

---

1. Black, K., Janner, M., Du, Y., Kostrikov, I., & Levine, S. (2023). Training Diffusion Models with Reinforcement Learning. ArXiv. https://arxiv.org/abs/2305.13301

**Questions:**

1. If the discriminator's objective is to specifically discriminate model outputs from PPD, how does it help to add the top-M ranking to positive labels? however I do acknowledge that using (y = e^{-α·r}) partially aleviates this concern. But from a theoretical standpoint, I do not understand how this could help, e.g., compared to only having the bottom-M added to the negative labels.
2. The dataset construction process is described in section 3.1. Are the first N raw outputs in $\mathcal{D}_r$ constant? the relation $\mathcal{D}=\mathcal{D}_r\bigcup\mathcal{D}_f^+\bigcup\mathcal{D}_f^-$ suggests that they are fixed. while $\mathcal{D}_f^\pm$ is updated. If that is the case, wouldn't periodically updating this set with new, challenging negatives from the improved generator be more effective?

---

### Official Review · Reviewer_9e4N · 2025-11-01

**Soundness:** 3
**Presentation:** 3
**Contribution:** 3
**Rating:** 4
**Confidence:** 4

**Summary:**

The paper proposes GAN-RM, a reward-modeling framework for visual generation that avoids human pairwise preference annotation and dimension engineering. It trains a discriminator to distinguish a small set of Preference Proxy Data (PPD) (hundreds of representative target samples) from model outputs, then uses scores for (i) best-of-N selection at inference and (ii) building datasets for SFT and DPO, with rank-based bootstrapping to iteratively improve both the reward model and the generator. Experiments on SD1.5, SDXL, and VideoCrafter2 show gains on FID, ImageReward, HPS, PickScore, CLIPScore, VBench, and safety metrics.

**Strengths:**

1. Reward modeling as discriminative prediction using PPD instead of labeled preferences.
2. Rank-based bootstrapping to expand pseudo labels and support multi-round post-training.
3. Demonstrations across image quality, safety alignment, and video generation, with competitive performance to DiffusionDPO trained on ~1M labels while using ~0.5k target samples.

**Weaknesses:**

1. GAN-RM can use only 0.5k data to achieve comparative performance with those methods that use 1M data. What about use 1M data to train GAN-RM? Table 11 is not enough to explain whether GAN-RM can be scaled.
2. Potential domain bias: Discriminator could learn style/domain artifacts of PPD rather than human “preference” per se; risk of reward hacking toward PPD distribution.

**Questions:**

1. Add experiments for scaling dataset size. It is a very important experiment for evaluation. Would like to raise the score if the experiment result is promising.

---

### Meta-Review · Area_Chair_a2d8 · 2025-12-27

**Summary:**

The consensus is that GAN-RM is fundamentally misaligned in its branding and technical execution. Reviewers noted that the "GAN" title is a misnomer, as the system lacks a true adversarial training loop, and they criticized the omission of text-conditioning, which leaves the reward model unable to judge if an image actually matches its prompt. Furthermore, the claim of removing human intervention is viewed as overstated, as the manual workload is simply shifted to the curation of "Preference Proxy Data" (PPD).

These conceptual gaps, coupled with the risk of reward hacking toward the proxy distribution, form the main concerns of the reviewers. Given the lack of a rebuttal and the negative consensus across reviewers, the AC recommends rejection of the paper.

**Reviewer Concerns:**

The consensus is that GAN-RM is fundamentally misaligned in its branding and technical execution. Reviewers noted that the "GAN" title is a misnomer, as the system lacks a true adversarial training loop, and they criticized the omission of text-conditioning, which leaves the reward model unable to judge if an image actually matches its prompt. Furthermore, the claim of removing human intervention is viewed as overstated, as the manual workload is simply shifted to the curation of "Preference Proxy Data" (PPD).

**Reviewer Scores:**

The initial scores would be kept as no rebuttal is posted.

---

### Decision · Program_Chairs · 2026-01-26

Reject